# Prevalence of COVID-19 and associated factors among healthcare workers in the war-torn Tigray, Ethiopia

Bisrat Tesfay Abera[1]*, Teklay Gebrecherkos[2], Migbnesh Gebremedhin Weledegebriel[1], Girmatsion Fisseha Abreha[3]

1 Department of Internal Medicine, School of Medicine, Mekelle University, Tigray, Ethiopia, 2 Department of Microbiology and Immunology, School of Biomedical Sciences, Mekelle University, Tigray, Ethiopia, 3 Department of Reproductive Health, School of Public Health, Mekelle University, Tigray, Ethiopia

* bis9live@gmail.com

**Data Availability Statement:** All relevant data are within the manuscript and the supporting information.

## Abstract

### Background

The Coronavirus disease 2019 (COVID-19) has put an enormous encumbrance on the healthcare system and healthcare workers (HCWs) worldwide, particularly in war-torn areas. As the world strives to end the pandemic, knowing the magnitude of the infection and its contributing factors in fragile settings is critical to prevent further waves of the pandemic.

### Methods

Using rapid diagnostic tests (RDTs), a facility-based cross-sectional study was conducted to assess the prevalence of COVID-19 and its associated factors. The study was conducted among 326 unvaccinated HCWs in two hospitals in Tigray, Ethiopia from. The study period was from June 26 to December 31, 2021. Descriptive statistics were used to analyze the characteristics of study participants and the magnitude of COVID-19 while multivariate logistic regression was applied to assess factors affecting COVID-19 infection among HCWs.

### Results

The seroprevalence of COVID-19 among HCWs in the war-torn region of Tigray was 52.4% and 56.5% using Cellex and INNOVA antibody RDTs, respectively. The point prevalence, using Abbot Antigen test, was 14.2%. The overall infection prevention and control practice (IPC) and facility preparedness was poor with >85% of the HCWs reporting not wearing eye goggle/face shield and respirator in activities that needed transmission-based precautions; none of the participants reporting as having separate waste disposal system for COVID-19 cases; and only 56.8% reporting as having an isolation area during the time of testing. In the multivariate analysis, not having isolation area (AOR = 19.6, 95% CI: 7.57–50.78), re-using of personal protective equipment (PPE) (AOR = 3.23, 95% CI: 1.54–6.77), being symptomatic (AOE = 2.4, 95% CI: 1.02–5.67), and being a medical doctor, doctor of dental surgery, and anesthetist (AOR = 3.64, 95% CI: 1.05–12.66) were significantly associated with having at least one positive result.

**Funding:** The author(s) received no specific funding for this work.

**Competing interests:** the authors have declared that no competing interests exist.

## Conclusions

Shortage of PPE supply, poor IPC practice, suboptimal facility preparedness, and low vaccination coverage in the region contributed to the high rate of COVID-19 infection among HCWs observed in this study.

## Introduction

As of October 18, 2022, severe acute respiratory distress syndrome virus 2 (SARS-CoV-2), which is the causative agent of Coronavirus disease 2019 (COVID-19), has resulted in 622,389,418 infections and 6,548,492 deaths globally since it was first reported from China on December 31, 2019 [1–4]. By October 18, 2022, there were around 12.661.050 COVID-19 cases and 257.749 deaths in Africa attributed to COVID-19 [5]. As of October 19, 2022, there have been 493,803 confirmed cases of COVID-19 and 7,572 deaths reported to WHO from Ethiopia [2].

Since the outbreak, Covid-19 has put an enormous encumbrance on the healthcare system and healthcare workers (HCWs) worldwide [6]. Their work characteristics exceptionally put them at a greater risk of acquiring the virus as they are at the frontline in fighting the outbreak. In a study done in the United Kingdom and the United States of America, there were 2,747 COVID-19 infected HCWs per 100,000, compared with 242/100,000 cases in the general public [7]. The number of health professionals who succumb to the disease so far is not to be underestimated either. According to the world health organization, about 80–180 thousand HCWs have perished due to the virus from January 2020 to May 2021 [8]. The brunt of the pandemic particularly falls on HCWs who live in war-torn and besieged areas. Breach in infection prevention and control (IPC) practices, lack of training, shortage of hygiene materials, poor facility preparedness, and breakdown of surveillance systems coupled with massive displacements put HCWs at a greater risk of COVID-19 [9]. War zones are usually excluded from programs and funds implemented to curb the pandemic worldwide. As a result, personal protective equipment (PPE), testing kits, and vaccines become inaccessible [8]. Studies done in Yemen have shown that COVID-19 and its related morbidity are high in war affected areas. Yemen has recorded a case fatality rate of 29% from COVID-19, which is the highest in the world. The seroprevalence of COVID-19 among HCWs is also high in the country [10].

COVID-19 mitigating programs and efforts started in Tigray, Ethiopia since the disease outbreak have been put in shambles as a result of a 2-year long war and siege [11]. Attacks on the health facilities and the health workforce have disintegrated the health system in Tigray putting an enormous obstacle in implementing COVID-19 preventive measures [12]. The war disrupted the existing COVID-19 surveillance measures and resulted in the closure of checkpoints and quarantine centers in the region [13]. No PPE and testing kit have made it to the region since the federal government and its allies imposed a brutal siege in June 2021 [14]. Like in Africa and other low and middle-income countries, including war-hit regions, vaccine coverage in Tigray remains critically low [15–17]. Blockades and war have meant Tigray has been left off from vaccination programs against COVID-19. Unpublished data from the region's health bureau showed that less than 2% of the population had received one dose of the AstraZeneca vaccine during the study period. This was followed by vaccination of only residents of Mekelle, the capital city of Tigray, in a 2-week campaign in July 2022. The plan was to vaccinate about 5 million of the population, but the campaign was halted since the resumption of the war in August 2022.

The war in Tigray has taken its toll on medical illnesses including infectious diseases. A study on the impact of the war on the service delivery of Human Immunodeficiency Virus disease revealed that the number of patients who had follow-up after 3 months of the start of the war dropped to 25% of the prewar period [18]. A Similar trend was seen in the care of patients with hypertension. The number of visits related to hypertension dropped by 52.7% during the first 8 months ensuing the war when compared to the pre-war period [19].

Although there have been concerns of increased transmission of the virus in Tigray since the eruption of the war, expressed by different bodies including UN agencies, the prevalence of the virus and its contributing factors among HCWs has not been studied during this cataclysmic period. A Myriad of factors including a breakdown in COVID-19 preventive measures, critically low vaccine coverage, and a dearth of supplies can put HCWs in the region at a considerable risk of acquiring the virus; hence increasing its prevalence among the HCWs in Tigray. Therefore, a facility-based cross-sectional study was conducted to assess the prevalence of COVID-19 among HCWs, using both antigen (Ag) and antibody (Ab) rapid diagnostic tests (RDTs), and its associated factors. This will help to see the burden of the problem for future COVID-19 prevention programs including vaccination.

## Methods and materials

### Study setting and study participants

The study was conducted in two hospitals (Ayder comprehensive specialized hospital (ACSH) and Mekelle general hospital) in Mekelle, Tigray, Ethiopia. Mekelle is the capital city of Tigray. ACSH and Mekelle Hospital give service to more than 9 million people with a catchment area involving Tigray, Eritrea, Afar, and northwestern parts of the Amhara region. The two hospitals have more than 700 in-patient beds in four major departments and other specialty units. The hospitals have the highest number of HCWs with 3000 and 500 in ACSH and Mekelle hospital, respectively. They also have the highest flow of patients and referrals among the hospitals in the Tigray region.

### Study design and sample size determination

A cross-sectional study design was used to determine COVID-19 positivity among unvaccinated HCWs in the two hospitals. The estimated sample size was 326. It was determined using 26.1% as the prevalence of COVID-19 among HCWs(taken from a Zimbabwean study[20]), 95% confidence interval, 5% margin of error, and 10% non-response rate. Data were collected from June 26 to December 30, 2021.

Information was obtained from the registries of each hospital to identify whether the HCWs were vaccinated against COVID-19 or not. There were 1,595 and 305 non-vaccinated HCWs in ACSH and Mekelle Hospital, respectively. The list of all non-vaccinated HCWs was identified to be included in the study and the vaccinated HCWs were excluded from the study. We excluded HCWs who were vaccinated to avoid false positives due to cross-reactivity with the vaccine [21]. Samples were then proportionally allocated among the two hospitals based on the number of unvaccinated HCWs (ACSH = (1,595/1,900)*317 = 274). Furthermore, the number of HCWs working at different units was considered for the proper allocation of samples at each unit in each facility. HCWs working at the COVID-19 isolation center, emergency outpatient department (OPD), intensive care unit (ICU), regular OPD, operation room (OR), and inpatient wards were selected from each hospital. After determining the allocated number of HCWs from each unit in each hospital, HCWs were randomly selected to participate in the current study (**Fig 1**).

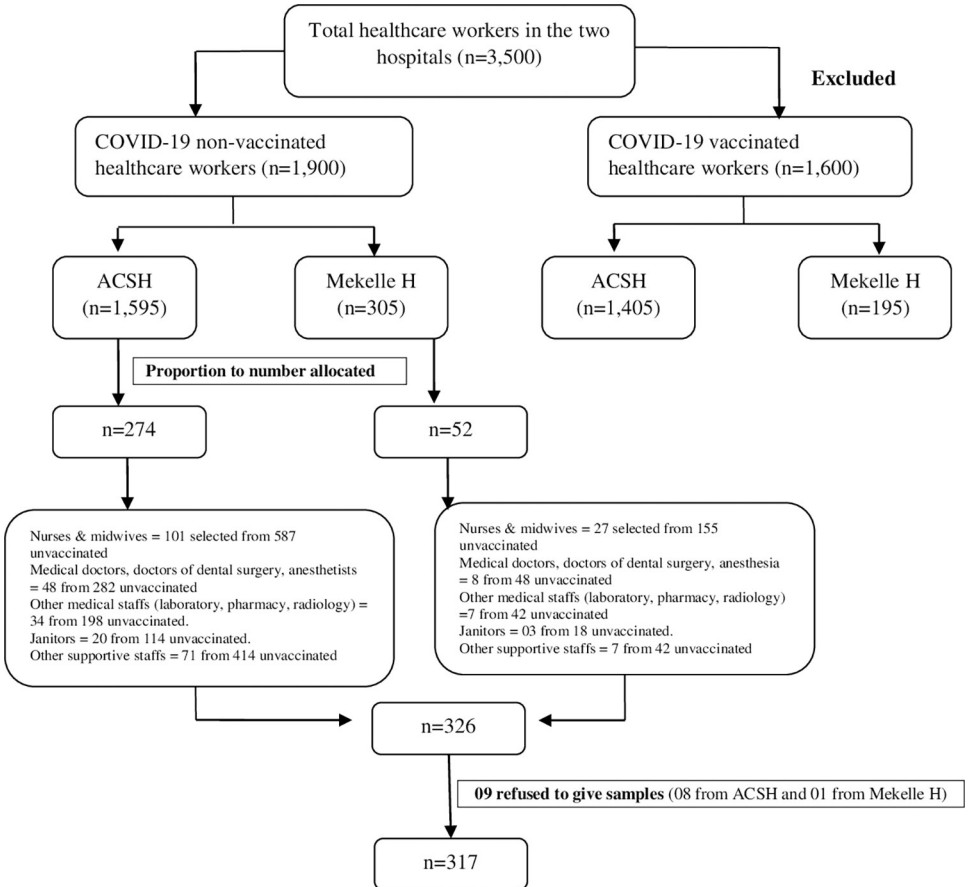

**Fig 1. Schematic presentation of sampling procedures used among healthcare workers, Tigray, Ethiopia, 2021.**

## Data collection tool and procedure

Data were collected using an interviewer-guided questionnaire, and taking blood and naso-pharyngeal samples from each of the selected HCWs. A structured questionnaire was used to collect data on socio-demographic characteristics, IPC practice, facility preparedness, and co-morbidities from the participants while a checklist was used to document the test results of three RDTs from the blood and nasopharyngeal samples. Two trained nurses and two labora-tory technicians collected the data. All the required instructions and techniques of sampling and testing were followed by trained laboratory technicians. Pre-test was done outside the selected hospitals before the data collection.

The RDTs of COVID-19 used in this study are described as follows. Pabbio $^{TM}$ COVID-19 Ag RDT (Abbott antigen rapid diagnostics Jena GmbH; orlaweg, Germany) was used for the qualitative detection of SARS-CoV-2 Ag in nasopharyngeal swab specimens of HCWs. **INNOVA** (Biological Technology; Beijing, China) lateral flow immune assay (LFIA) and **Cellex** LFIA which target the nuclear (N) and spike (S) proteins of SARS-CoV-2 and are designed to detect IgM and IgG against SARS-CoV-2 were used to detect the presence of antibodies in blood samples. Testing was undertaken following the manufacturer's instructions for the assays. In all the RDTs, result interpretation was based on the appearance of colored bands; interpreted as positive when control and test bands were visible; as negative when only the control band was visible; or as invalid when the control band was not visible. All results were

read by two technologists independently. There was strict supervision during data collection, blood and nasopharyngeal sampling, and testing of results.

## Measurements

The outcome variable of this study was COVID-19 using the different RDTs. Socio-demographic characteristics of HCWs, IPC practice, facility preparedness, and co-morbidities were used as explanatory variables.

**Acute/recent covid-19 in serologic tests:** When the result was IgM positive or reactive; that is when only the IgM line developed in addition to the presence of the C line.

**Previous/recent covid-19 in serologic tests:** When the result was IgG positive or reactive; that is when only the IgG line developed in addition to the presence of the C line.

**At least one positive test result:** having a positive result for Cellex (Cel) IgM and/or Cel IgG and/or INNOVA IgM and/or IgG and/or Ag.

**Healthcare workers**: all paid and unpaid individuals who work in the healthcare setting who are at risk of both direct and indirect exposure to COVID-19 infected patients or their infectious secretions and materials (e.g., doctors, nurses, laboratory workers, facility or maintenance workers, cleaners, security guards, clinical trainees, volunteers, etc.).

**Aerosol-generating procedures (AGPs)**: are procedures done in a hospital setting that generate aerosols when performed in patients with respiratory illnesses such as COVID-19. The following procedures are usually considered AGPs (Intubation and extubation; Manual ventilation; Open suctioning; Cardiopulmonary resuscitation; Bronchoscopy and ENT fibreoptic endoscopy (unless carried out through a closed-circuit ventilation system); Dental procedures; Non-invasive ventilation; Continuous positive airway pressure ventilation; High-frequency oscillatory ventilation; Induction of sputum).

## Data analysis

Descriptive statistics were used to analyze the characteristics of study participants and the magnitude of COVID-19 while bivariate and multivariate logistic regression was applied to assess the association between dependent and independent variables. Pearson Chi-square test was also used to see the relationship between variables. Variables with a p-value < 0.2 in the bivariate analysis were chosen to be analyzed in multivariate analysis to minimize the effect of confounding factors between variables. The presence and strength of association were determined using an odds ratio with a 95% confidence interval. Variables with a P value < 0.05 were assumed as statistically significant. Hosmer-Lemeshow goodness of fit test was run which showed that our model was a good model for the fitted variables.

## Ethical consideration

Ethical clearance was obtained from the Ethical Review Committee of Mekelle University College of health sciences' institutional review board (IRB) with an IRB number of 1863/2021. Participants were briefed on the purpose of the study and informed written consent was obtained from each participant. Additional written consent was taken for blood and nasopharyngeal samples. Anonymity was assured to the study participants. The authors did not have access to individual information.

## Results

Of the 326 eligible HCWs, a total of 317 HCWs were included in the study, making the response rate 97.2%. Most of the study participants (184 (58.0%)) were in the age group

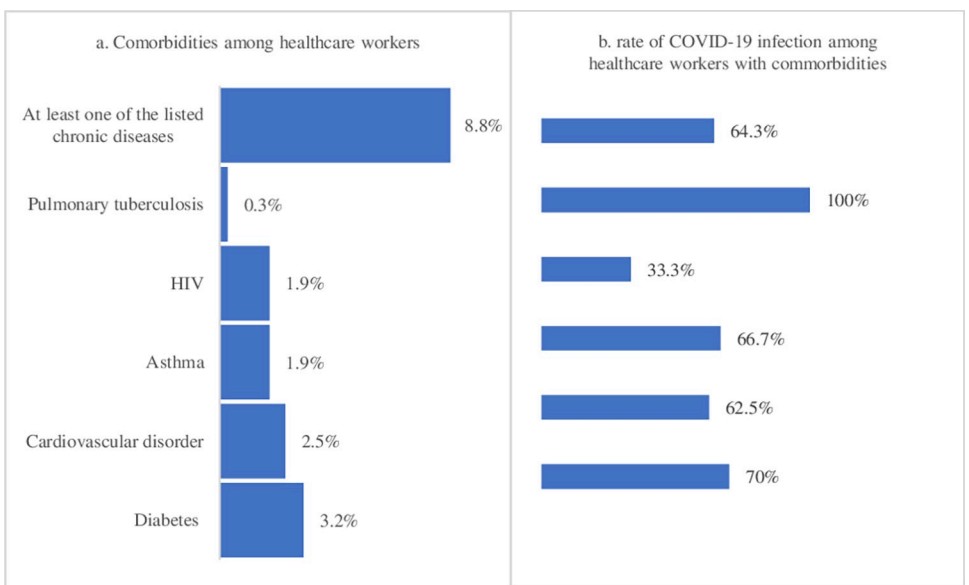

**Fig 2. Comorbidity status and risk of COVID-19 among healthcare workers in Tigray, Ethiopia, 2021(n = 317).**

between 25 and 36 years, and the average age was 32 ±8.4 years. Above half of the participants (186 (58.7%)) were female and married (167 (52.7%)). Nurses by profession accounted to 35.6% (113) while medical doctors (MD) and supportive staff accounted to 13.6% (43) and 31.0% (98) of the study participants, respectively (**S1 Table in S1 File**).

## Comorbidity status of study participants

Of the total participants, 06 (1.9%), 10 (3.2%), and 08 (2.5%) HCWs reported having HIV, diabetics mellitus, and cardiovascular disorder (Chronic rheumatic valvular heart disease and hypertension), respectively. Overall, 28 (8.8%) of the study participants had at least one comorbidity. The rate of COVID-19 was 70%, 67%, and 63% among HCWs with diabetes, Asthma, and cardiovascular disorder, respectively (**Fig 2**).

## Magnitude of COVID-19 among healthcare workers in Tigray

Three RDTs were used to assess the status of COVID-19 among the HCWs. There was variation among diagnostic tests in detecting the COVID-19 infection. The seroprevalence of COVID-19 among HCWs was 52.4% (166) and 56.5% (179) using Cellex and INNOVA Ab RDTs, respectively. The rate of acute/recent infection using INNOVA and Cellex Ab RDTs was 22.7% (72) and 6% (19), respectively. A total of 150 (47.3%) and 159 (50.2%) HCWs were positive for INNOVA IgG and Cel IgG, respectively. The point prevalence of COVID-19 using Abbott Ag test was 14.2% (45). Overall, 207 (65.3%) of the HCWs had at least one positive result, indicating a very high prevalence of COVID-19 among HCWs in the health facilities included in the study. The rate of infection was higher among males than females in Ab RDTs, but it was not statistically (P-value>0.05) (**Table 1**).

## Relationship between COVID-19 test positivity and symptoms

The Majority of COVID-19-positive HCWs tested using Cel IgM (84.2%), Ag (91.1%), and INNOVA IgM (48.6%) manifested with symptoms of COVID-19 (p<0.001). The test results from Cel IgG and INNOVA IgG were not related to showing symptoms (**Table 2**).

**Table 1. COVID-19 rapid diagnostic test results among healthcare workers in Tigray, Ethiopia, 2021 (n = 317).**

| Rapid diagnostic test | Female (n = 186) | | Male (n = 131) | | Total | |
|---|---|---|---|---|---|---|
| | Frequency | Percentage | Frequency | Percentage | Frequency | Percentage |
| **INNOVA IgM** | | | | | | |
| Positive | 39 | 21.0% | 33 | 25.2% | 72 | 22.7 |
| Negative | 147 | 79.0% | 98 | 74.8% | 245 | 77.3 |
| **INNOVA IgG** | | | | | | |
| Positive | 85 | 45.7% | 65 | 49.6% | 150 | 47.3 |
| Negative | 101 | 54.3% | 66 | 50.4% | 167 | 52.7 |
| **Cel IgM** | | | | | | |
| Positive | 8 | 4.3% | 11 | 8.4% | 19 | 6.0 |
| Negative | 178 | 95.7% | 120 | 91.6% | 298 | 94.0 |
| **Cel IgG** | | | | | | |
| Positive | 92 | 49.5% | 67 | 51.1% | 159 | 50.2 |
| Negative | 94 | 50.5% | 64 | 48.9% | 158 | 49.8 |
| **Abbot RDT (Ag test)** | | | | | | |
| Positive | 28 | 15.1% | 17 | 13.0% | 45 | 14.2 |
| Negative | 158 | 84.9% | 114 | 87.0% | 272 | 85.8 |
| **At least one positive result of the five** | | | | | | |
| Yes | 119 | 64.0% | 88 | 67.2% | 207 | 65.3 |
| No | 67 | 36.0% | 43 | 32.8% | 110 | 34.7 |
| **Cel IgG and/or Cel IgM** | | | | | | |
| Positive | 97 | 52.2% | 69 | 52.7% | 166 | 52.4 |
| Negative | 89 | 47.8% | 62 | 47.3% | 151 | 47.6 |
| **INNOVA IgG and/or INNOVA IgM** | | | | | | |
| Positive | 100 | 53.8% | 79 | 60.3% | 179 | 56.5 |
| Negative | 86 | 46.2% | 52 | 39.7% | 138 | 43.5 |
| **Acute/recent COVID-19 (INNOVA IgM and/or Cel IgM)** | | | | | | |
| Positive | 39 | 21.0% | 33 | 25.2% | 72 | 22.7 |
| Negative | 147 | 79.0% | 98 | 74.8% | 245 | 77.3 |
| **Acute/recent COVID-19 (INNOVA IgM and/or Cel IgM and/or Ag)** | | | | | | |
| Positive | 54 | 29.0% | 40 | 30.5% | 94 | 29.7 |
| Negative | 132 | 71.0% | 91 | 69.5% | 223 | 70.3 |
| **Previous/recent COVID-19 (INNOVA IgG and/or Cel IgG)** | | | | | | |
| Positive | 97 | 52.2% | 71 | 54.2% | 168 | 53.0 |
| Negative | 89 | 47.8% | 60 | 45.8% | 149 | 47.0 |

## Working areas, contact status, and related risk to COVID-19

Of the total study participants, 24 (7.5%) participants were working in the COVID-19 isolation center, 98 (30.9%) in the emergency room, and 65 (20.5%) in the regular OPD at the time of testing. The probability of having one positive test result was high in those who worked in the emergency OPD, but it was not statistically significant. On average, most of the participants worked for eight hours per day. Only 23 (7.3%) study participants worked for more than eight hours per day. About 289 (91.2%) participants said that they did not have separate donning, doffing, utility, and clean storage rooms in their working unit at the time of testing. More than one-third (110 (34.7%)) of participants reported a history of contact with COVID-19 confirmed or suspected cases within 14 days of testing. Contact with patients was the most commonly reported type of contact (91(82.7%)) **(Table 3)**.

**Table 2. Relationship between positive test results and symptoms of COVID-19 among healthcare workers in Tigray, Ethiopia, 2021 (n = 317).**

| Test result | | Symptoms of COVID-19 within 14 days of testing | | Total | χ2 | p-value |
|---|---|---|---|---|---|---|
| | | Yes | No | | | |
| **Cel IgM** | +ve | 16 (84.2%) | 3 (15.8%) | 19 (100%) | 30.975 | <**0.001** |
| | -ve | 74 (24.8%) | 224 (75.2%) | 298 (100%) | Ref. | Ref. |
| **Cel IgG** | +ve | 44 (27.7%) | 115 (72.3%) | 159 (100%) | 0.081 | 0.776 |
| | -ve | 46 (29.1%) | 112 (70.9%) | 158 (100%) | Ref. | Ref. |
| **Ag** | +ve | 41 (91.1%) | 4 (8.9%) | 45 (100%) | 101.476 | <**0.001** |
| | -ve | 49 (18.0%) | 223 (82.0%) | 272 (100%) | Ref. | Ref. |
| **INNOVA IgM** | +ve | 35 (48.6%) | 37 (51.4%) | 72 (100%) | 18.734 | <**0.001** |
| | -ve | 55 (22.4%) | 190 (77.6%) | 245 (100%) | Ref. | Ref. |
| **INNOVA IgG** | +ve | 42 (28.0%) | 108 (72.0%) | 150 (100%) | 0.021 | 0.884 |
| | -ve | 48 (28.7%) | 119 (71.3%) | 167 (100%) | Ref. | Ref. |

p-value using Pearson Chi-square test.

## COVID-19 prevention and control practice

Less than half (131 (41.3%)) of study participants received IPC training. About 283 (89.3%), 277 (87.4%), and 131 (41.3%) HCWs reported using surgical mask, practicing hand hygiene, and wearing gloves, respectively, as part of standard precautions. Only few workers of the health facilities practiced respiratory hygiene (50 (15.3%)) and used eye wear (19(6.0%)) as part of standard precautions. About 307 (86.8%) and 113 (35.6%) used surgical mask and glove/utility glove, respectively, when performing activities that needed transmission-based precautions. Only 22 (6.9%), 21 (6.6%), 44 (13.9%) wore eye goggle/face shield, long sleeved gown/Coverall, and Respirator (N95/FFP3), respectively. From the 29 (9.1%) HCWs who performed aerosol generating procedures within 14 days of testing, no one reported as wearing a respirator during the procedures. Most (191 (60.3%)) of the HCWs re-used PPE in their daily activities. Medical mask/surgical mask (184 (96.3%)), respirator (N95/FFP3) (22 (11.5%)), and glove/utility glove (10 (5.2%)) were the most commonly re-used PPE.

From the total participants, 37 (11.7%), 119 (37.5%), 148 (46.8%), and 47 (14.8%) reported practicing social distancing, wearing mask, performing hand hygiene, and performing respiratory hygiene, respectively, in the community (when outside of the health facility) (**S2 Table in S1 File**).

## Health facility preparedness to prevent COVID-19

Study participants were asked if their hospital had isolation center for contact, suspected, and confirmed COVID-19 cases at the time of testing, and 180 (56.8%) of them responded positively. Only 131 (41.3%) reported that their facility had triaging system for COVID-19 cases at the time of testing. All HCWs in this study perceived that their health facilities did not have separate laundry area and waste disposal system for COVID-19 cases. There was no separate residence for HCWs who worked in areas isolated for COVID-19 quarantined, suspected, and confirmed cases. Few (71 (22.4%)) HCWs reported that their units offered masks to quarantined, suspected or confirmed COVID-19 patients. Very few HCWs attested having IPC team (12 (3.8%)) and environmental health team (14 (4.4%)) in their unit (**Table 4**).

**Table 3. Working environment, status of contact and associated risk to COVID-19 among healthcare workers in Tigray, Ethiopia, 2021(n = 317).**

| Working environment | Frequency | Percentage | At least one positive result | p-value[b] |
|---|---|---|---|---|
| **Working area** | | | | |
| COVID-19 isolation center | 24 | 7.6 | 13(54.2%) | 0.012 |
| Emergency | 98 | 30.9 | 77 (78.6%) | |
| ICU | 12 | 3.8 | 09 (75%) | |
| Regular OPD | 65 | 20.5 | 42 (64.6%) | |
| Operation room | 22 | 6.9 | 14 (63.6%) | |
| Ward | 96 | 30.3 | 52 (54.2%) | |
| **Average working hour per day** | 8 hours (SD:1.7) | | | |
| **Average working hour per day** | | | | |
| < = 8hrs | 294 | 92.7 | 190 (64.6%) | 0.254 |
| >8hrs | 23 | 7.3 | 17 (73.9%) | |
| **History of contact [a]** | | | | |
| Yes | 110 | 34.7 | 82 (74.5%) | 0.008 |
| No | 207 | 65.3 | 125 (60.4%) | |
| **Contact with COVID-19 confirmed cases** | | | | |
| Yes | 58 | 52.7 | 47 (81.0%) | 0.076 |
| No | 52 | 47.3 | 35 (67.3%) | |
| **Contact with COVID-19 suspected cases** | | | | |
| Yes | 69 | 62.7 | 47 (68.1%) | 0.035 |
| No | 41 | 37.3 | 35 (85.4%) | |
| **Contact with cases who had contact with COVID-19 confirmed or suspected cases** | | | | |
| Yes | 3 | 2.7 | 3 (100%) | 0.410 |
| No | 107 | 97.3 | 79 (73.8%) | |
| **The person contacted** | | | | |
| Patient | 91 | 82.7 | 65 (71.4%) | 0.082 |
| Colleague | 53 | 48.2 | 39 (73.6%) | 0.498 |
| Friend | 38 | 34.5 | 27 (71.1%) | 0.348 |
| Family member | 16 | 14.5 | 11 (68.8%) | 0.383 |

a = having contact with COVID-19 confirmed or suspected cases within 2-meter distance for more than 15 minutes within 14 days of testing.

b = p-value using Pearson chi-square or fisher's Exact test.

## Factors associated with COVID-19

In the multivariate analysis, not having an isolation area for contacts, suspected, and confirmed COVID-19 cases in the health facility (AOR = **19.6, 95% CI: 7.57–50.78**), reusing of PPE (AOR = **3.23, 95% CI: 1.54–6.77**), having symptoms of COVID-19 within 14 days of testing (AOE = **2.4, 95% CI: 1.02–5.67**)**,** and being an MD, doctor of dental surgery (DDS), and anesthetist (AOR = **3.64, 95% CI: 1.05–12.66**) were significantly associated with having at least one positive result (**Table 5**).

Moreover, having a history of contact with suspected or confirmed COVID-19 individuals within 14 days of testing (AOR = **2.96, 95% CI: 1.09–8.0**) was significantly associated with being positive for Abbott Ag test while receiving IPC training (AOR = **0.24, 95% CI: 0.07–0.85**) conferred protection (**S4 Table in S1 File**). Wearing a mask when out in the community was protective against having Cel IgM and/or INNOVA IgM positive results (AOR = **0.32, 95% CI: 0.15–0.68**) (**S5 Table in S1 File**). In addition, age 35–44 (AOR = **2.69, 95% CI: 1.46–4.93**) and age>45 (AOR = **3.31, 95% 1.04–10.54**) were significantly associated with having INNOVA IgG and/or Cel IgG positive results while working in units that offer masks to

**Table 4. Health facility preparedness to tackle COVID-19 in Tigray, Ethiopia, 2021 (n = 317).**

| Facility preparedness | Frequency | % |
|---|---|---|
| **Having isolation area during testing** | | |
| Yes | 180 | 56.8 |
| No | 137 | 43.2 |
| **Separate residence for HCWs at isolation areas** | | |
| Yes | 0 | 0 |
| No | 317 | 100 |
| **Separate laundry area for COVID-19 cases** | | |
| Yes | 0 | 0 |
| No | 317 | 100 |
| **Separate waste disposal system for COVID-19 cases** | | |
| Yes | 0 | 0 |
| No | 317 | 100 |
| **Triaging system for COVID-19 cases during testing** | | |
| Yes | 131 | 41.3 |
| No | 186 | 58.7 |
| **Unit offers masks to quarantined, suspected or confirmed COVID-19 cases** | | |
| Yes | 71 | 22.4 |
| No | 246 | 77.6 |
| **IPC team in the unit** | | |
| Yes | 12 | 3.8 |
| No | 305 | 96.2 |
| **Environmental health team in the unit** | | |
| Yes | 14 | 4.4 |
| No | 303 | 95.6 |

quarantined, suspected, or confirmed COVID-19 cases (AOR = **0.31, 95% CI: 0.15–0.67**) was protective (**S6 Table in S1 File**).

## Discussion

By conducting a survey using both Ab and Ag RDTs, we found that the rate of COVID-19 among HCWs in the war-wracked region of Tigray was high across all tests; and there was variation in the prevalence of the infection using the different tests. The concordance rate between symptoms and test positivity in the RDTs that detect active SARS-CoV-2 infection also showed disparity, with INNOVA IgM having the lowest concordance rate. Four sets of factors were associated with SARS-CoV-2 infection among HCWs in the current study. These findings shed light on how HCWs practicing in fragile regions, where disruption of the health sector is ubiquitous, are affected by the COVID-19 pandemic. Factors related to sociodemographic characteristics, IPC practice, facility preparedness, exposure risk, and clinical features contributed to the increased prevalence of infection with SARS-COV-2 among HCWs in the current study. These results discern contributing elements that need intervention to halt COVID-19 transmission in the health facilities in particular and in the community at large in war-torn regions, as high prevalence in health facilities can be a snapshot of what is happening in the community, and as the virus can gain a foothold in these regions paving the way for future waves of the pandemic.

The prevalence of COVID-19 among HCWs was high compared to studies in other parts of the world. The seroprevalence of COVID-19 among HCWs in most developed countries

**Table 5. Factors associated with having at least one positive result to COVID-19 rapid diagnostic tests among healthcare workers, Tigray, Ethiopia, 2021 (n = 317).**

| Variables | At least one positive result | | COR (95%, CI) | AOR (95%, CI) |
|---|---|---|---|---|
| | Yes | No | | |
| **Age in years** | | | | |
| < = 24 | 22 | 18 | 1 | 1 |
| 25–34 | 120 | 64 | 1.53 (0.77, 3.07) | 2.17 (0.74, 6.38) |
| 35–44 | 46 | 20 | 1.88 (0.83, 4.25) | 3.02 (0.85,10.76) |
| > = 45 | 19 | 08 | 1.94 (0.69,5.47) | 2.21 (0.47, 10.36) |
| **IPC training** | | | | |
| Yes | 46 | 71 | 1 | 1 |
| No | 161 | 39 | **6.37 (3.83, 10.61)**\*** | 1.59 (0.74, 3.42) |
| **Isolation area** | | | | |
| Yes | 55 | 102 | 1 | 1 |
| No | 152 | 08 | **35.2 (16.1, 77.09)**\*** | **19.6 (7.57, 50.78)**\*** |
| **Reusing PPE** | | | | |
| Yes | 167 | 35 | **8.95 (5.27, 15.19)**\*** | **3.23 (1.54, 6.77)**\** |
| No | 40 | 75 | **1** | **1** |
| **Contact with suspected and/or confirmed COVID-19 cases** | | | | |
| Yes | 82 | 28 | **1.92(1.15, 3.2)**\** | 0.84 (0.37, 1.83) |
| No | 125 | 82 | **1** | 1 |
| **Wearing mask in the community** | | | | |
| Yes | 56 | 63 | **0.28 (0.17, 0.45)**\*** | 0.66 (0.31, 1.39) |
| No | 151 | 47 | **1** | 1 |
| **Symptoms of COVID-19 within 14 days of testing** | | | | |
| Yes | 73 | 17 | **2.98(1.65, 5.38)**\*** | **2.4 (1.02, 5.67)**\* |
| No | 134 | 93 | **1** | **1** |
| **Working hours per day** | | | | |
| < = 8hrs | 190 | 104 | 0.65 (0.25, 1.69) | 4.46 (0.79, 25.39) |
| >8hrs | 17 | 06 | 1 | 1 |
| **Working unit during the time of testing** | | | | |
| COVID-19 isolation center | 13 | 11 | 1 (0.41, 2.45) | 3.32 (0.96, 11.41) |
| EM and regular OPDs | 119 | 44 | **2.29 (1.35, 3.89)**\*** | 0.96, 5.59) |
| ICU and OR | 23 | 11 | 1.77 (0.78, 4.03) | 1.61 (0.46, 5.68) |
| Wards | 52 | 44 | 1 | 1 |
| **Occupation** | | | | |
| Nurse or Midwife | 83 | 41 | 1.5 (0.72, 3.11) | (0.36, 4.02) |
| MD, DDS, ANT [a] | 44 | 11 | **2.96 (1.19, 7.35)**\* | **3.64 (1.05, 12.66)**\* |
| Other Health professionals [b] | 23 | 17 | 1 | 1 |
| Supportive staff (Janitors) | 9 | 13 | 0.51 (0.18, 1.47) | 0.64 (0.12, 3.39) |
| Other supportive staffs [c] | 48 | 28 | 1.27 (0.58, 2.77) | 1.46 (0.46, 4.68) |

\*p-value <0.05

\**p-value 0.01–0.002

\***p-value < = 0.001.

a = Medical doctor, Doctor of Dental surgery, Anesthetist

b = Laboratory technician/technologist, pharmacist, radiology technician/technologist

c = Casher, security, morgue attendant, porter, oxygen attendant, social worker, maintenance worker, hospital data encoder.

ranges from 1% to 24.4%, much lower than the rate of infection reported in the present study [22]. The infection rate in our study is also higher than the rate of COVID-19 among HCWs reported from developing nations which reaches up to 45.1% [23]. High seroprevalence of COVID-19 among HCWs is also witnessed in Yemen, a country ravaged by war since 2014. A study in one of the country's hospitals revealed a seroprevalence of 19.4% [10]. Although both settings share similarities, the complete siege on Tigray during the study period might have contributed to the much higher prevalence observed in our study. The study in Yemen was carried out early in the pandemic, thus an additional reason for the relatively low seroprevalence of COVID-19 among Yemeni HCWs when compared to our study. An array of reasons is responsible for the increased COVID-19 rate among HCWs observed in our setting and other conflict-affected areas where COVID-19 mitigating programs have collapsed because of war and siege. Failures in IPC practice and training, breakdown of surveillance and triaging systems, poor facility preparedness, lack of testing, low vaccination coverage, and shortage of PPE tied to the conflict and blockade are the likely reasons why HCWs in Tigray are exposed to the virus to this extent [13]. A high prevalence of COVID-19 among HCWs will result in an increased risk of intrahospital transmission and threaten patients' lives [24]. The incidence of COVID-19 among HCWs in the current study is higher than the incidence of COVID-19 in the general population in Tigray. A study done before the war broke out showed a 4.3% incidence of COVID-19 in the general population [25]. Although it is difficult to draw a conclusion as the two studies were done under different circumstances and used separate testing modalities, the finding complies with previous studies which indicated an increased risk of COVID-19 among HCWs compared to the general population. Their profession demands working in propinquity with infected individuals which puts them at a higher risk of contracting the virus when compared to the general population, especially when they are under-protected due to a lack of PPE and other preventive measures [26].

The difference in the rate of detection of the infection, and the variation in the test positivity and symptom concordance rate seen among the different RDTs is supportive of findings by D. S.Y. Ong et al. who reported variability in the rate of detection of SARS-CoV-2 infection among the different Ag and Ab testing lateral flow devices [27]. The variability in sensitivity and specificity of the tests, viral load, population level prevalence, and the presence or absence of symptoms and their timing can elucidate the findings observed in the studies [28]. Therefore, though rapid tests have an acceptable performance, other factors should also be considered when interpreting the results of the RDTs [29].

Among the sociodemographic characteristics associated with increased risk of infection were older age in both IgG tests and occupation (MD, DDS, and anesthetist) in the Abbot rapid Ag detecting test. Older age was significantly associated with acquiring the disease in a study done by Sophie Alice Muller et al. in their scoping review of COVID-19 seroprevalence and its risk factors in HCWs in Africa [23], but in contrast to findings in most studies carried out in the developed world which showed no significant association between age and the risk of contracting the virus [6,30,31]. The reason can be because of gaps in knowledge about the virus and its preventive measures and differences in practice and adherence to IPC strategies. Old-aged HCWs in Africa may perform less in these parameters [32]. This finding calls for stepping up of training related to COVID-19 preventive modalities, increasing PPE supply, and improving vaccine coverage, particularly in the older HCWs who are at an increased risk of morbidity and mortality due to the virus [33]. The increased risk of infection among MDs, DDS, and anesthetists is similar to some studies done during the early stage of the pandemic, which showed physicians and HCWs exposed to the airway and oral cavity to be at a high risk of the infection [31,34], but it is contradictory to most studies which reported high prevalence of the SARS-CoV-2 infection among nurses and other HCWs [35–37]. The low compliance to

IPC programs and shortage of IPC materials due to the war and blockade are pervasive and affect all HCWs in our setting. As a result, other factors such as the presence or absence of contact with COVID-19 suspected or confirmed individuals will play a critical role in the acquisition of the virus [7], which is the case in our study where having a history of contact with suspected or confirmed patients was highest among physicians, DDS, and anesthetists **(S3 Table in S1 File).** The finding underscores the need for giving special attention to those who have frequent exposure to patients when considering deploying mitigating programs in resource limited areas.

The overall IPC practice and facility preparedness were poor in this study, mirroring the dire situation war-affected regions are facing during the pandemic [9]. Not having IPC training, PPE reuse, working in units that do not offer masks to patients, and not having an isolation area were significantly associated with a high risk of SARS-CoV-2 infection among the HCWs in Tigray. Similar findings were observed in other studies in which PPE reuse and inadequate IPC training were associated with an increased risk of contracting the virus [7,34,38]. The breach in IPC practice and inadequate facility preparedness creates a fertile ground for virus transmission in health facilities, thereby putting HCWs at an increased risk of acquiring the virus [32]. Poor compliance with COVID-19 mitigating strategies of health facilities in war-shattered and beleaguered zones not only affects the HCWs, but puts patients in particular and the community in general at risk as the facilities will continue to be a habitat for the virus [39]. This underpins the need for integrated support from different bodies, including governmental and local and international non-governmental institutions, to control the transmission of the virus in war-torn regions. If they are precluded from the mitigating programs (including vaccination) employed by international organizations such as the world health organization to contain the virus, they can be a breeding ground for the virus and become a niche for outbreaks and future waves of the pandemic [17].

Having contact with COVID-19 suspects or confirmed patients within 14 days of testing was another factor significantly associated with having a positive test in the RDTs that detect active SARS-Cov-2 infection. The increased risk of infection when having a history of contact with COVID-19 patients is further confirmed by Ran L et al. and Nugyen et al. [7,40]. This is owing to inadequate protection of the HCWs when coming into contact with patients, largely due to a shortage of PPE supply and lack of IPC training which is particularly evident in fragile settings [41]. On the contrary, Abdulla A. et al. reported no association between direct patient contact and risk of infection [42]. This can be explained by the difference in the degree of PPE supplies, adherence to recommendations, and capacity building of HCWs in different settings, with facilities in war zones expected to be much less prepared [43]. Our study did not find any difference in the risk of infection among health units. This contrasts studies by Adrian Shields et al. and Jeardon Nevardo Malagón-Rojas et al. which indicated that working in an intensive care unit conferred protection more than working in other areas [22,44]. This is because facilities usually channel their available resources, including PPE and training, to high-risk areas [44]. But this is not true for health institutions in embattled areas where COVID-19 preventive measures are non-existent [45]. Therefore, HCWs across all units are left with no option but to give care to patients not armed with adequate resources and knowledge, again making nosocomial transmission and outbreaks very likely.

Finally, being symptomatic was significantly associated with having positive results in the tests that determine acute infection. This aligns with studies by Shields A et al. and Kassem AK et al. which showed that the presence of symptoms was significantly associated with having a positive test result [30,44]. Test positivity increases during symptomatic periods when there is an increased viral shading [46]. Not having a proper triaging system to trace and track individuals with symptoms and the absence of isolation centers, which is the case in our study, can

result in the uncontrollable spread of the infection within facilities [47]. With constraints of testing modalities during a humanitarian crisis in consideration, priority should be given to those who showed symptoms when considering testing and isolating individuals.

## Strengths and limitations of the study

This study tried to assess the prevalence of COVID-19 and its associated factors among HCWs amid the humanitarian crisis in Tigray. It has tried to bring to light how disruption of the health system and low vaccination coverage can increase COVID-19 in war-ravaged areas. The study used quantitative (questionnaire) and qualitative (blood and nasopharyngeal samples) data using three standard RDTs that gave a picture of recent and previous COVID-19 among HCWs. However, it has limitations. Since nucleic acid amplification tests were inaccessible in the region during the study period, we could not assess the specificity and sensitivity of the RDTs. In addition, we included only unvaccinated HCWs. As a result, we could not see the effect of COVID-19 vaccines on COVID-19 infection in our setting.

## Conclusion

In this study, the prevalence of COVID-19 among HCWs was among the highest in the world. The inability to provide training to HCWs on the preventive measures of COVID-19, shortage of PPE supply, difficulty in sustaining facility preparedness at the level required, and low vaccination coverage in the region have contributed to the high prevalence of the infection observed during this catastrophic period. Therefore, we recommend that the responsible bodies, including international organizations, increase the supply of PPE, give trainings on COVID-19 preventive measures, provide vaccines, and help reestablish triaging systems and isolation centers in the studied facilities in particular and in the region in general.

## Supporting information

**S1 Checklist.**
(DOCX)

**S2 Checklist. STROBE statement—Checklist of items that should be included in reports of observational studies.**
(DOCX)

**S1 File.**
(DOCX)

## Acknowledgments

All authors are grateful for the support received from ACSH and for the study participants who consented to be included in the study. We also acknowledge the data collectors who accomplished their tasks efficiently without receiving their due payment due to the blockade of the bank system in Tigray, Ethiopia.

## Author Contributions

**Conceptualization:** Bisrat Tesfay Abera.

**Data curation:** Bisrat Tesfay Abera, Teklay Gebrecherkos, Girmatsion Fisseha Abreha.

**Formal analysis:** Bisrat Tesfay Abera, Girmatsion Fisseha Abreha.

**Investigation:** Bisrat Tesfay Abera, Teklay Gebrecherkos.

**Methodology:** Bisrat Tesfay Abera, Teklay Gebrecherkos, Girmatsion Fisseha Abreha.

**Project administration:** Bisrat Tesfay Abera.

**Software:** Girmatsion Fisseha Abreha.

**Supervision:** Bisrat Tesfay Abera, Teklay Gebrecherkos.

**Validation:** Bisrat Tesfay Abera, Teklay Gebrecherkos, Migbnesh Gebremedhin Weledegebriel, Girmatsion Fisseha Abreha.

**Writing – original draft:** Bisrat Tesfay Abera, Girmatsion Fisseha Abreha.

**Writing – review & editing:** Bisrat Tesfay Abera, Teklay Gebrecherkos, Migbnesh Gebremedhin Weledegebriel, Girmatsion Fisseha Abreha.

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
