## [Decision Letter · Decision Letter 0]

15 May 2023

PONE-D-22-34484

Impact of war on COVID-19 infection among healthcare workers in Tigray, Northern Ethiopia

PLOS ONE

Dear Dr. Abera,

Thank you for submitting your manuscript to PLOS ONE. After careful consideration, we feel that it has merit but does not fully meet PLOS ONE’s publication criteria as it currently stands. Therefore, we invite you to submit a revised version of the manuscript that addresses the points raised during the review process.

We look forward to receiving your revised manuscript.

Kind regards,

Werku Etafa

Academic Editor

PLOS ONE

Additional Editor Comments:

1.     You submitted research entitled as “Impact of war on COVID-19 infection among healthcare workers in Tigray, Northern Ethiopia”. The study was conducted using a cross-sectional study design which lasted about five months. The assessment of impact requires data for a long period of years. However, in your study, you haven’t mentioned anything about the effects of COVID-19 on HCWs. It begins and ends with prevalence and its associated factors. Besides, there is no single question that is targeted to assess the effect/impact. Thus, you are recommended to modify your title “Prevalence of COVID-19 and associated factors among HCWs in Tigray, Ethiopia” as it could fit to your findings.  

2.     It is essential to differentiate whether it was a war or conflict in Tigray region during the study period. Is there formally declared for war at the time of your data collection? Sensitive issues, seek evidence. Thus, you are requested to modify with the available evidence.

3.     COVID-19 is an infectious respiratory disease. In the acronym, COVID-19, “I” stands for Infectious. Thus, no need to use it repeatedly.

4.     When you mention authors' affiliation, it is suggested to state from the specific unit to the general unit (For instance: department …, school/college, institute/…, university, region, country). Do not intermix.

5.     The corresponding author is the main author who communicates by the journal regarding your work. You mentioned all of your members as the corresponding author as seen in the first page of your manuscript. Please try to adhere to the journal guideline.

“We are grateful for the unwavering support we have received from Tigray regional Health bureau and Ayder Comprehensive Specialized Hospital. Tigray regional health bureau supported the study with data collection fee, diagnostic tests and personal protective equipment. We also acknowledge the data collectors who accomplished their task efficiently without receiving their due payment due to the blockade of bank system in Tigray, Ethiopia.”

“All authors have no conflict of interest.”

Reviewers' comments:

Reviewer's Responses to Questions

**Comments to the Author**

1. Is the manuscript technically sound, and do the data support the conclusions?

Reviewer #1: Partly

2. Has the statistical analysis been performed appropriately and rigorously? 

Reviewer #1: Yes

3. Have the authors made all data underlying the findings in their manuscript fully available?

Reviewer #1: Yes

4. Is the manuscript presented in an intelligible fashion and written in standard English?

Reviewer #1: Yes

5. Review Comments to the Author

Reviewer #1: The authors revealed an important findings. If the following points are included, the paper will contribute the covid-19 literature among health workers in conflict setting.

See the comments below.

Abstract

• Line 30—revise the statement “ ….. the association between dependent and independent variables” to “…. factors affecting COVID-19 infection among health workers”.

• Line 31-- Results: while the title mentions conflict, I don’t see any key word stating conflict. Any idea for the authors on this issue?

Introduction

• Lines 80-86: Any data that conflict causes extra covid-19 infection, mortality and other related complications?

• Line 90: citations needed

• Line 95—remove the subtitle ‘local context’, and cut and paste the points about local vaccine coverage from lines 105-11 to after line 95.

• NB. Could the authors make ‘Tigrai/y’ consistent? Perhaps use Tigray throughout?

• In rationale of the study, the authors need to put their hypothesis and contextualized to conflict. For example, they could describe how the war in Tigray caused increased incidence and prevalence of illnesses (e.g. HIV infection), mortality and loss-to follow up (e.g. hypertension), and others. And covid-19 infection could also be hypothesized that there incidence among HWs will be high. This way it can be contextualized to the conflict.

Methods

• Line 123: Why are study participants unvaccinated? I think it’s important to consider the case in all health workers so that even vaccine effectiveness among health workers could be described. Could the authors put their rationale why they only included mention unvaccinated health works in the method section but also add a limitation at the end of discussion that they could have included vaccinated HWs and even saw how the effectiveness of the vaccine among those vaccinated/unvaccinated?

Results

• Line 332/3: I wonder why the CI is very high for the variable suspected and confirmed COVID-19 cases in the health facility (AOR=19.6, 95% CI: 7.57, 50.78).

• While the result is presented very well, I still wonder how the title was contextualized to the conflict in Tigray --- the authors need to add some contextualized concept.

Discussion

• Line 377-379. I think the comparison should also be done with other conflict-affected areas

• Could the authors compare if the incidence among the health workers was different when compared to the general population in the region? And how was the incidence when compared to the incidence before the start of the conflict in the region?

• Add the above mentioned limitation

NB. The paper needs heavy English proof.

6. PLOS authors have the option to publish the peer review history of their article (what does this mean?). If published, this will include your full peer review and any attached files.

Reviewer #1: **Yes: **Hailay Abrha Gesesew

---

## [Author Response · Author response to Decision Letter 0]

25 May 2023

Dear editors and reviewers 

First of all, we would like to thank you very much for the chance you gave us to revise the manuscript. We are very pleased to know that our manuscript was rated as potentially acceptable for publication in PLOS, subject to a radical revision and response to the comments raised by the editor and the reviewer.

Based on the instructions provided in the journal's website and your email on May 15, 2023, we attached the file of our revised manuscript word (doc.). We have revised the manuscript along the line of all comments made by the editor and the referee. 

Appended to this letter is our point-by-point response to the comments raised by the referee and the editor. 

Response to editor and reviewer

A. Comments from the editor 

1. You submitted research entitled as “Impact of war on COVID-19 infection among healthcare workers in Tigray, Northern Ethiopia”. The study was conducted using a cross-sectional study design which lasted about five months. The assessment of impact requires data for a long period of years. However, in your study, you haven’t mentioned anything about the effects of COVID-19 on HCWs. It begins and ends with prevalence and its associated factors. Besides, there is no single question that is targeted to assess the effect/impact. Thus, you are recommended to modify your title “Prevalence of COVID-19 and associated factors among HCWs in Tigray, Ethiopia” as it could fit to your findings. 

Response:

Thank you very much for your invaluable comments. We accept your suggestion and we have changed the title to “Prevalence of COVID-19 and associated factors among HCWs in the war-torn Tigray, Ethiopia.” Please see the title in the revised manuscript with track changes. We are open for further suggestions. 

2. It is essential to differentiate whether it was war or conflict in Tigray region during the study period. Is there formally declared for war at the time of your data collection? Sensitive issues seek evidence. Thus, you are requested to modify with the available evidence. 

Response:

Thank you for raising this comment. The data was collected during the second phase of the fighting where the war had already spilled over to other regions of Ethiopia. According to a book by Dan smith, war is defined as “an armed conflict involving at least two or more centrally organized parties, in disputes about power over government and territory, engaging in armed conflict in formal military formation that have caused at least the death of 1000 people.”(https://berghof-foundation.org/library/trends-and-causes-of-armed-conflict) The fight betweenTigray region on the one end and the Federal government of Ethiopia, regional special forces and Eritrea on the other end has resulted in the death of 100s of thousands of people.In addition, different institutional bodies, including the European parliament, have addressed the situation as war. (https://news.un.org/en/story/2022/09/1127481, https://www.europarl.europa.eu/thinktank/en/document/EPRS_BRI(2022)739244. 

We are open for further comments and suggestions.

3. COVID-19 is an infectious respiratory disease. In the acronym, COVID-19, “I” stands for Infectious. Thus, no need to use it repeatedly. 

Response:

Thank you again. We have removed it from statements where we put it next to “COVID-19”. Please see the amendment in the revised manuscript with track changes.

4. When you mention authors affiliation it is suggested to state from the specific unit to the general unit (For instance: department of…, school/college, institute/…, university, region, country). Do not intermix. 

Response:

Thank you for this constructive comment. We have put the authors’ affiliations according to your suggestion. Please see the amendment in the revised manuscript with track changes.

5. The corresponding author is the main author who communicate with the journal regarding your work. You mentioned all of your members as the corresponding author as seen in the first page of your manuscript. Please try to adhere to the journal guideline. 

Response:

Thank you for this constructive comment. The corresponding author of this manuscript is Dr. Bisrat Tesfay Abera. We have removed the other authors from the corresponding author list. Please see the amendment in the revised manuscript with track changes.

6. English language usage still need great effort. 

Response:

Thank you again for your constructive comment.

To the best of our knowledge, we have tried to improve the language usage. Please see the amendment in the revised manuscript with track changes.

7. The scientific reasonings in the discussion section and the methods needs revision. 

Response:

Thank you again for your constructive comment.

To the best of our knowledge, we have tried to revise the methods and discussion. Please see the amendment in the revised manuscript with track changes.

B. Reviewer comments

8. Abstract

• Line 30—revise the statement “ ….. the association between dependent and independent variables” to “…. factors affecting COVID-19 infection among health workers”.

Response:

Thank you for the invaluable and constructive comments. We have changed the statement as per your suggestion. Please see the revision in the updated manuscript with track changes.

• Line 31-- Results: while the title mentions conflict, I don’t see any key word stating conflict. Any idea for the authors on this issue?

Response:

Thank you for raising this concern. As per the editor’s suggestion, we agreed to change the title and removed the “impact of the war” from the title. But, to not to completely ignore the contribution of the war, we have added the phrase “in the war-torn region of Tigray.” We are ready for further suggestions and comments. Please see the change we have made in the revised manuscript with track changes.

Introduction

• Lines 80-86: Any data that conflict causes extra covid-19 infection, mortality and other related complications?

Response:

Thank you for the suggestion again. We have incorporated studies from Yemen. Please see the revision made in the manuscript with track changes.

• Line 90: citations needed

Response:

Thank you for your input. We accept your suggestion and we have added a citation. Please see the revision in the manuscript with track changes.

• Line 95—remove the subtitle ‘local context’, and cut and paste the points about local vaccine coverage from lines 105-11 to after line 95.

Response:

Thank you for your suggestion. We have removed the subtitle and moved the content about the vaccine coverage in Tigray to the recommended part of the introduction. please look in the manuscript with the track changes.

• NB. Could the authors make ‘Tigrai/y’ consistent? Perhaps use Tigray throughout?

Response: 

We acknowledge your comment and have tried to be consistent with the word Tigray.

• In rationale of the study, the authors need to put their hypothesis and contextualized to conflict. For example, they could describe how the war in Tigray caused increased incidence and prevalence of illnesses (e.g. HIV infection), mortality and loss-to follow up (e.g. hypertension), and others. And covid-19 infection could also be hypothesized that theincidence among HWs will be high. This way it can be contextualized to the conflict.

Response:

Thank you again for this important insight. We have added the effect of the war on HIV service delivery and care of patients with hypertension in Tigray, and to the best of our knowledge, we have tried to contextualize it to the conflict. Please see the revised manuscript with track changes.

• Line 123: Why are study participants unvaccinated? I think it’s important to consider the case in all health workers so that even vaccine effectiveness among health workers could be described. Could the authors put their rationale why they only included mention unvaccinated health works in the method section but also add a limitation at the end of discussion that they could have included vaccinated HWs and even saw how the effectiveness of the vaccine among those vaccinated/unvaccinated?

Response:

Thank you again. Since we used serology tests to assess the prevalence of COVID-19 among healthcare workers, we excluded HCWs who were vaccinated to avoid cross reactivity and minimize false positivity. We accept your suggestion and have included the reasons why we only considered the unvaccinated HCWs in the methods section. We have also considered it as a limitation for not including the vaccinated HCWs at the end of the discussion section. Please see the manuscript with track changes. (Considerations for the use of antibody tests for SARS CoV-2 - first update (europa.eu))

Results

• Line 332/3: I wonder why the CI is very high for the variable suspected and confirmed COVID-19 cases in the health facility (AOR=19.6, 95% CI: 7.57, 50.78).

Response:

Thank you for the comment. In table 5 of the result section, the variable “Contact with suspected and/or confirmed COVID-19 case” is not significant at multivariate analysis, only significant at bivariate analysis where there was higher risk of infection among HCWs who had contact with suspected/confirmed COVID-19 case (COR=1.92, 95% CI: 1.15, 3.2). The high CIs was for the variable “Isolation area” (AOR=19.6, 95% CI: 7.57, 50.78). This indicates those HCWs who work outside the COVID-19 isolation area were more likely to experience COVID-19 infection. HCWs who work at isolation center might have more awareness about the disease and might have received more training compared to those HCWs working in other units. Regarding the high CI; it could be due to the small sample size in the cells; of the four cells, one cell has 08 samples. This may lead to wide CI and higher odds ratio. 

• While the result is presented very well, I still wonder how the title was contextualized to the conflict in Tigray --- the authors need to add some contextualized concept.

Response:

Thank you for this critical input. As mentioned above, we have revised the title as we did not have data to compare the difference in prevalence of COVID-19 before the start of the war and during the war. We are open for further suggestions and comments.

Discussion

• Line 377-379. I think the comparison should also be done with other conflict-affected areas

Response:

Thank you again. We accept the recommendation and we have added data from Yemen. Please see the revision in the revised manuscript with track changes.

• Could the authors compare if the incidence among the health workers was different when compared to the general population in the region? And how was the incidence when compared to the incidence before the start of the conflict in the region?

Response:

Thank you for this recommendation. We have managed to get data about the incidence of the infection in the general population before the war and we have included it in the discussion. Unfortunately, we were not able to find data about the incidence of COVID-19 in the general population during the war. Please see the revision in the revised manuscript with track changes.

• Add the above mentioned limitation

Response:

Thank you very much. We accept your suggestion and we have put it as one of the limitations of the study. Please see the revision in the revised manuscript with track changes.

NB. The paper needs heavy English proof.

Response:

Thank you very much. We accept the recommendation and have done the English proofing. Please see the manuscript with track changes.

Sincerely,

Bisrat, MD, Internist.

---

## [Decision Letter · Decision Letter 1]

14 May 2024

PONE-D-22-34484R1Prevalence of COVID-19 and associated factors among healthcare workers in the war-torn Tigray, Ethiopia.PLOS ONE

Dear Dr. Abera,

Thank you for submitting your manuscript to PLOS ONE and patiently waiting through the peer review process. After careful consideration, we feel that it has merit but does not fully meet PLOS ONE’s publication criteria as it currently stands. Therefore, we invite you to submit a revised version of the manuscript that addresses the points raised during the review process. The manuscript still has significant grammatical errors and can be improved by using a grammar software like Grammarly. In addition, review the tables as was suggested by reviewer 2. 

Please submit your revised manuscript within Jun 28 2024 11:59PM. If you will need more time than this to complete your revisions, please reply to this message or contact the journal office at plosone@plos.org. Please include the following items when submitting your revised manuscript:A rebuttal letter that responds to each point raised by the academic editor and reviewer(s). You should upload this letter as a separate file labeled 'Response to Reviewers'.A marked-up copy of your manuscript that highlights changes made to the original version. You should upload this as a separate file labeled 'Revised Manuscript with Track Changes'.An unmarked version of your revised paper without tracked changes. You should upload this as a separate file labeled 'Manuscript'.If applicable, we recommend that you deposit your laboratory protocols in protocols.io to enhance the reproducibility of your results. Protocols.io assigns your protocol its own identifier (DOI) so that it can be cited independently in the future. For instructions see: https://journals.plos.org/plosone/s/submission-guidelines#loc-laboratory-protocols. Additionally, PLOS ONE offers an option for publishing peer-reviewed Lab Protocol articles, which describe protocols hosted on protocols.io. Read more information on sharing protocols at https://plos.org/protocols?utm_medium=editorial-email&utm_source=authorletters&utm_campaign=protocols.

We look forward to receiving your revised manuscript.

Kind regards,

Ogochukwu Chinedum Okoye

Academic Editor

PLOS ONE

Journal Requirements:

Additional Editor Comments:

Reviewers' comments:

Reviewer's Responses to Questions

**Comments to the Author**

1. If the authors have adequately addressed your comments raised in a previous round of review and you feel that this manuscript is now acceptable for publication, you may indicate that here to bypass the “Comments to the Author” section, enter your conflict of interest statement in the “Confidential to Editor” section, and submit your "Accept" recommendation.

Reviewer #2: (No Response)

Reviewer #3: (No Response)

2. Is the manuscript technically sound, and do the data support the conclusions?

Reviewer #2: Yes

Reviewer #3: Yes

3. Has the statistical analysis been performed appropriately and rigorously? 

Reviewer #2: Yes

Reviewer #3: Yes

4. Have the authors made all data underlying the findings in their manuscript fully available?

Reviewer #2: Yes

Reviewer #3: Yes

5. Is the manuscript presented in an intelligible fashion and written in standard English?

Reviewer #2: Yes

Reviewer #3: Yes

6. Review Comments to the Author

Reviewer #2: REVIEW OF ARTICLE

TITLE : Prevalence of COVID-19 and associated factors among healthcare workers in the war-torn Tigray, Ethiopia.

Abstract

Results

Line 38- 40 : The confidence Interval should be written in form of range. Example, 7.57- 50.78

Introduction

Page 4,line 91: … existing COVID-19 surveillance NOT existed COVID-19 surveillance

Methodology

Page 6, line 118: can be revised to “ The hospitals have more than 700 in-patient beds…” rather than “The hospitals have more than 700 beds inpatients beds …”

Data analysis

Page 9, line 192: Hosmer-Lemeshow goodness of fit test was run .. √

Hosmer-Lemeshow goodness of fittest was ran… X

Results

It is important that the author should start by briefly telling us how the 97.2% response rate was arrived at.

Page 10, line 206, should be written as “ … average age was 32.0 ± 8.4 years.”

Discussion

Page 20,line 385: Insert” likely” between the and reasons = the likely reasons …

Page 20, line 398: this finding is in line with what… √

This finding is in line to what … X

Page 22, line 445

This could be explained by the difference in degree … √

This could be explianed by the deference in degree … X

References

Line 513, reference 6: Lapolla PMA √

Lapolla P, M.A. X

Line 520, reference 8: There should be full-stop after the author’s name, that is, after WHO.

The days for which the literature was cited online should be written.

Line 569, reference 24: Insert fullstop after the name of author, ie CDC.

Line 588, reference 30: There should be up to 6 authors spelt out before stating et al.

Reviewer #3: I thank the PLOS team for the opportunity to review this manuscript. I was privy to the first reviewer's comments and the response of the manuscript corresponding author (when I received this article for review the first time). A lot of the issues I would have raised have already been raised by an initial reviewer and addressed by the corresponding author.

However a few further observations and corrections to be done include;

Conclusion:

1. Please rephrase this statement "escalate pieces of training on COVID-19 preventive measures" ('escalate' as a word in that context seems inappropriate)

Tables and Figures:

2. In the tables S3, S4 and S5. there were no percentages (in brackets) seen besides their respective counts/frequencies. Conventionally contingency tables should be presented first descriptively before being reported inferential or predictive. The authors should quickly rectify this. Kindly compare them to table S6 which was better formatted/done (there will be no need for the total column in S3,S4 and S5). Figure 3 is better understood as a table.

7. PLOS authors have the option to publish the peer review history of their article (what does this mean?). If published, this will include your full peer review and any attached files.

Reviewer #2: No

Reviewer #3: **Yes: **Awunor, Nyemike Simeon

---

## [Author Response · Author response to Decision Letter 1]

23 May 2024

First of all, we would like to thank you very much for the chance you have given us to revise the manuscript. We are very pleased to know that our manuscript was rated as potentially acceptable for publication in PLOS ONE, subject to revision and response to the comments raised by the editor and the reviewers.

Based on the instructions provided in the journal's website and your email on May 14, 2024, we have attached the file of our revised manuscript with track changes, the clean version and response to reviewers. We have revised the manuscript along the line of all comments made by the editor and reviewers. 

Appended to this letter is our point-by-point response to the comments raised by reviewers.

Sincerely,

Bisrat, MD, Internist. 

Mekelle University, College of Health Sciences, Tigray, Ethiopia.

Response to editor and reviewers

A. Comments from Reviewer 1

1. Abstract

Results: Line 38- 40: The confidence Interval should be written in form of range. Example, 7.57- 50.78 

Response:

Thank you very much for your invaluable comments! We accept your suggestion and we have modified it accordingly. Please see the title in the revised manuscript with track changes. 

2. Introduction

Page 4,line 91: … existing COVID-19 surveillance NOT existed COVID-19 surveillance

Response:

Thank you! We have rearranged the sentence according to your suggestion. Please see the title in the revised manuscript with track changes. 

3. Methodology

Page 6, line 118: can be revised to “ The hospitals have more than 700 in-patient beds…” rather than “The hospitals have more than 700 beds inpatients beds …”

Response:

Thank you for this constructive comment. We have amended the sentence according to your suggestion. Please see the amendment in the revised manuscript with track changes.

4. Data analysis

Page 9, line 192: Hosmer-Lemeshow goodness of fit test was run .. √

Hosmer-Lemeshow goodness of fittest was ran… X 

Response:

Thank you again! We have revised it in line to your suggestion. Please see the amendment in the revised manuscript with track changes.

5. Results

It is important that the author should start by briefly telling us how the 97.2% response rate was arrived at.

Page 10, line 206, should be written as “ … average age was 32.0 ± 8.4 years.” 

Response:

Thank you again for your constructive comment! We have included a sentence about how the 97.2% was arrived. We have also corrected line 206 according to your suggestion. Please see the amendment in the revised manuscript with track changes.

6. Discussion

Page 20, line 385: Insert” likely” between the and reasons = the likely reasons …

Page 20, line 398: this finding is in line with what… √

This finding is in line to what … X

Page 22, line 445

This could be explained by the difference in degree … √

This could be explianed by the deference in degree … X

 Response:

Thank you again! We have revised them in line to your suggestions. However, we could not get this line: “this finding is in line with what “ in the final manuscript that was submitted after the previous revision Please see the amendment in the revised manuscript with track changes.

7. References

Line 513, reference 6: Lapolla PMA √

Lapolla P, M.A. X

Line 520, reference 8: There should be full-stop after the author’s name, that is, after WHO. The days for which the literature was cited online should be written.

Line 569, reference 24: Insert full stop after the name of author, ie CDC.

Line 588, reference 30: There should be up to 6 authors spelt out before stating et al.

Response:

Thank you again for your constructive comment. We have revised them according to your suggestions. However, Line 24 is moved to line 31 in the final manuscript that was submitted after the first revision and we have put the full stop after CDC in line 31. We have also included the days the literature was cited/accessed. In the previously revised manuscript, we had listed 6 authors before et al in all references that had more than 6 authors. Please see the amendment in the revised manuscript with track changes.

B. Comments from Reviewer 2

1. Conclusion

Please rephrase this statement "escalate pieces of training on COVID-19 preventive measures" ('escalate' as a word in that context seems inappropriate)

Tables and Figures:

Response:

Thank you again for your constructive comment. We have revised it according to your suggestions. Please see the amendment in the revised manuscript with track changes.

2. Tables and figures

In the tables S3, S4 and S5. there were no percentages (in brackets) seen besides their respective counts/frequencies. Conventionally contingency tables should be presented first descriptively before being reported inferential or predictive. The authors should quickly rectify this. Kindly compare them to table S6 which was better formatted/done (there will be no need for the total column in S3,S4 and S5). Figure 3 is better understood as a table.

 Response:

Thank you again for your constructive comment. We have revised supplement tables and figure according to your suggestions. Please see the amendment in the revised manuscript with track changes.

---

## [Decision Letter · Decision Letter 2]

26 Aug 2024

Prevalence of COVID-19 and associated factors among healthcare workers in the war-torn Tigray, Ethiopia.

PONE-D-22-34484R2

Dear Dr. Bisrat,

We’re pleased to inform you that your manuscript has been judged scientifically suitable for publication and will be formally accepted for publication once it meets all outstanding technical requirements.

Kind regards,

Ogochukwu Chinedum Okoye

Academic Editor

PLOS ONE

Additional Editor Comments (optional):

Reviewers' comments:

Reviewer's Responses to Questions

**Comments to the Author**

1. If the authors have adequately addressed your comments raised in a previous round of review and you feel that this manuscript is now acceptable for publication, you may indicate that here to bypass the “Comments to the Author” section, enter your conflict of interest statement in the “Confidential to Editor” section, and submit your "Accept" recommendation.

Reviewer #3: All comments have been addressed

2. Is the manuscript technically sound, and do the data support the conclusions?

Reviewer #3: Yes

3. Has the statistical analysis been performed appropriately and rigorously? 

Reviewer #3: Yes

4. Have the authors made all data underlying the findings in their manuscript fully available?

Reviewer #3: Yes

5. Is the manuscript presented in an intelligible fashion and written in standard English?

Reviewer #3: Yes

6. Review Comments to the Author

Reviewer #3: The article is technically sound, the data analysis is appropriate, the manuscript is written in an intelligible fashion and the standard of English is much improved.

My comments in review 1 has been addressed by the authors.

7. PLOS authors have the option to publish the peer review history of their article (what does this mean?). If published, this will include your full peer review and any attached files.

Reviewer #3: No

---

## [Editor Report · Acceptance letter]

29 Aug 2024

PONE-D-22-34484R2 

PLOS ONE

Dear Dr. Abera, 

I'm pleased to inform you that your manuscript has been deemed suitable for publication in PLOS ONE. Congratulations! Your manuscript is now being handed over to our production team.

Kind regards, 

on behalf of

Dr. Ogochukwu Chinedum Okoye 

Academic Editor

PLOS ONE